# Novel Method to Detect Pitfalls of Intracoronary Pressure Measurements by Pressure Waveform Analysis

**DOI:** 10.3390/jpm12122035

**Published:** 2022-12-08

**Authors:** Csaba Jenei, Balázs Tar, András Ágoston, Péter Sánta, János Sánta, Benjámin Csippa, Richárd Wéber, Dániel Gyürki, Gábor Halász, Gábor Tamás Szabó, Dániel Czuriga, Zsolt Kőszegi

**Affiliations:** 1Division of Cardiology, Department of Cardiology, Faculty of Medicine, University of Debrecen, 4032 Debrecen, Hungary; 2Kálmán Laki Doctoral School of Biomedical and Clinical Sciences, University of Debrecen, 4032 Debrecen, Hungary; 3Szabolcs—Szatmár—Bereg County Hospitals and University Teaching Hospital, 4400 Nyíregyháza, Hungary; 4Semicolon Ltd., 1083 Budapest, Hungary; 5Department of Hydrodynamic Systems, Budapest University of Technology and Economics, 1111 Budapest, Hungary

**Keywords:** fractional flow reserve measurement, hyperemic pressure gradient, technical pitfall of FFR measurement, pressure signal drifting

## Abstract

Potential pitfalls of fractional flow reserve (FFR) measurements are well-known drawbacks of invasive physiology measurement, e.g., significant drift of the distal pressure trace may lead to the misclassification of stenoses. Thus, a simultaneous waveform analysis of the pressure traces may be of help in the quality control of these measurements by online detection of such artefacts as the drift or the wedging of the catheter. In the current study, we analysed the intracoronary pressure waveform with a dedicated program. In 130 patients, 232 FFR measurements were performed and derivative pressure curves were calculated. Local amplitude around the dicrotic notch was calculated from the distal intracoronary pressure traces (δdP_n_/dt). A unidimensional arterial network model of blood flow was employed to simulate the intracoronary pressure traces at different flow rates. There was a strong correlation between δdP_n_/dt values measured during hyperaemia and FFR (r = 0.88). Diagnostic performance of distal δdP_n_/dt ≤ 3.52 for the prediction of FFR ≤ 0.80 was 91%. The correlation between the pressure gradient and the corresponding δdP_n_/dt values obtained from all measurements independently of the physiological phase was also significant (r = 0.80). During simulation, the effect of flow rate on δdP_n_/dt further supported the close correlation between the pressure ratios and δdP_n_/dt. Discordance between the FFR and the δdP_n_/dt can be used as an indicator of possible technical problems of FFR measurements. Hence, an online calculation of the δdP_n_/dt may be helpful in avoiding some pitfalls of FFR evaluation.

## 1. Introduction

Coronary angiography is a gold standard technique for assessing coronary artery disease (CAD). In the majority of cases, percutaneous coronary intervention (PCI) relies on information derived from coronary angiography. However, the prognosis of CAD is only improved when PCI targets a coronary artery stenosis responsible for the ischemia [1]. Importantly, the physiological significance of a coronary artery stenosis often cannot be fully determined by angiography alone. This is of utmost importance when the angiographic severity of the stenosis is intermediate (50–70% diameter stenosis). International guidelines support the use of physiological measurements to assess the functional severity of coronary lesions [2,3]. Evidence indicates that the fractional flow reserve (FFR) determined by intracoronary pressure measurement provides reliable information concerning the hemodynamic significance of a coronary lesion [4]. An FFR value below 0.80 implies significance, and it characterizes a stenosis that can induce myocardial ischemia (physiologically or functionally significant coronary lesion). Despite the fact that the FFR measurement may improve the appropriateness of the decision for revascularization, the use of FFR is still restricted to a small portion of intermediate coronary lesions [5].

The FFR measurement is readily reproducible with a quite small variability [6,7]. However, in several cases, the operator may encounter potential technical pitfalls during the examination. The drift of the pressure signal is related to the catheter used and it can easily produce misleading results [8]. Pressure signal drift can be recognized when comparing the aortic (proximal) and distal (intracoronary) pressure signals.

The drift mainly arises from the technical properties of the piezo resistive pressure wire sensor. However, this phenomenon was also observed when using fiberoptic sensors. Inevitably, all pressure wire devices have some level of measurement inaccuracy. According to literature data, the acceptable drift should be lower than 5 mmHg/h [8,9]. If the signal drift is higher than 5 mmHg during the procedure, the measurement should be repeated to avoid any misinterpretation of the physiological assessment of the stenosis. One problem is that the drift may go unnoticed during the measurement, until the pressure wire is completely withdrawn into the guiding catheter at the end of the procedure. Some level of drift is always considered possible; hence, verification of the pressure signals is crucial [8,10]. The pressure drift can be calculated by subtracting the mean coronary artery pressure (Pd) from the mean aortic pressure (Pa) obtained following the pressure wire transducer pullback maneuver.

Before a comprehensive evaluation of pressure signal traces, the real impact and incidence of pressure wire drifts were unknown. Recently, a study revealed that clinically accepted degrees of pressure wire drift result in stenosis misclassification in up to one-third of cases [10].

Another technical pitfall of the pressure wire measurement is related to the wedging of the coronary catheter tip in the ostium. This results in a damping of the aortic pressure signal while also disturbing the distal pressure trace.

There is no official guideline for an effective management of all technical problems of FFR determination; however, the early detection of the pressure signal drift or the wedging of the catheter may improve the quality of FFR measurement. Both problems can be resolved by analyzing the pressure waveform, and the morphological characteristic of the dicrotic notch seems a proper tool for it [9].

The present study seeks to demonstrate the feasibility of performing a pressure waveform analysis by focusing on the dicrotic notch using dedicated software for the verification of FFR measurement accuracy based on the distal pressure waveform and the actual pressure gradient.

## 2. Materials and Methods

Study design. We studied 130 consecutive patients (62 ± 10 years of age) with a wide range of suspected or known coronary artery diseases, referred for invasive investigation to the Hemodynamic laboratory of the Department of Cardiology and Cardiac Surgery, University of Debrecen. All patients underwent diagnostic coronary angiography indicated mainly for stable angina. Patient data (e.g., previous medical history) were stored in the hospital information system, while procedural data were stored in a dedicated local PACS database and analyzed retrospectively.

In the catheterization laboratory, mostly radial coronary angiographies were performed, but the femoral access was also used at the operator’s discretion in a minority of cases where the radial approach was not feasible. The coronary angiographies were performed with a 6 Fr diagnostic or guiding catheter, and the angiographic recordings were digitally stored in an archive system (GE Maclab). All patients signed a written informed consent, and the study protocol was approved by the local ethics committee.

When the diagnostic examination revealed a lesion of angiographically intermediate degree (50–70% diameter stenosis) without documented ischemia on non-invasive tests, the physician performed an intracoronary FFR measurement to assess the functional severity of the coronary artery stenosis. The aortic pressure transducer was attached to the table at a reference height of 5 cm below the sternum. While the guiding catheter was positioned in the coronary ostium, the pressure wire (PressureWire Certus, Radi Medical Systems/ St. Jude) was fully flushed with room temperature saline, and then it was zeroed and calibrated according to the manufacturer’s instruction. Pressure equalization was performed with the pressure wire sensor positioned at the tip of the guiding catheter. Next, the pressure wire was advanced, and the sensor was positioned distally to the stenosis. The FFR measurements were acquired during intracoronary (ic) bolus injections of adenosine (100–200 μg ic). During the procedure, the proximal (intra-aortic) and distal (intracoronary) pressure curves were simultaneously recorded. The recording was continuous throughout the entire hyperaemic phase until the adenosine effect completely disappeared (resting phase). The maximum hyperaemic, steady-state pressures were used to calculate the FFR, while resting pressures were used to determine the resting pressure ratio (RPR) with the RadiAnalyser equipment (Radi Medical Systems, Uppsala, Sweden). FFR was determined as the ratio of the mean coronary artery pressure distal to the stenosis over the pressure measured proximally at the tip of the guiding catheter during maximum hyperaemia. At the end of the recording, the pressure sensor was withdrawn into the catheter to identify possible pressure drifts. If the pressure drift was more than 5 mmHg, the recording was discarded, and the measurement was repeated. The stenosis was identified as a functionally severe or ischemia-provoking lesion if the measured FFR was <0.80.

After the procedure, the pressure curves were exported from the RadiAnalyser equipment through the RadiView 2.0 software package to our dedicated, custom-made JAVA-based software to calculate the derivative curves. Both RadiView and our JAVA-based software used the same time resolution; hence, cursor locations identified exactly the same positions where FFR or RPR measurements had been performed earlier.

The dedicated, JAVA-based software works in three steps (Figure 1):The operator placed the cursor just before the dicrotic notch of the original pressure trace.The software automatically determined the local amplitude (the difference between the local maximum and the minimum value) around the dicrotic notch using the derivative curves of the pressure traces (δdP_n_/dt).The process was repeated for three consecutive cardiac cycles adjacent to the selected time-point, and the average values were calculated and used for further analysis.

### 2.1. A Unidimensional Numerical Model of Blood Flow

In addition, a mathematical model was used to simulate the intracoronary flow in a certain hemodynamic condition. To examine the effect of the flow rate on δdp_n_/dt, simulations were made using a unidimensional model of blood flow at different flow rates. The model was based on a previously published numerical method [11] that can be used to simulate blood flow in arterial networks. In this model, blood is treated as a Newtonian fluid and the viscoelastic properties of the vessel wall are also taken into account by the so-called Stuart model. A detailed description of the model was published earlier [12].

At the inlet, the aortic flow (p_a_) was prescribed as a periodic boundary condition, which could be obtained from the measurement. At the outlet, the pressure was calculated based on the left ventricular pressure (p_LV_) using the varying elastance model [13].

In our model, the stenosis and the arterioles were represented by two resistors connected in series (Figure 2), the values of which were calculated based on the distal pressure (p_d_) and the flow rate. The original flow rate was 4.66 mL/s. Simulations were performed at different flow rates from 0% to 175% of the original value.

### 2.2. Statistical Methods

All the statistical analyses were performed with SPSS 14.0 for Windows (Statistical Product and Service Solutions, version 14, SPSS Inc., Chicago, IL, USA). Normality was assessed with a normal probability (Q–Q) plot and with a non-parametric Kolmogorov–Smirnov test. All continuous variables were reported as the mean ± standard deviation. Also, analyses were performed with the Student’s *t*-test for continuous data. Non-normal distributed values were expressed as the median (interquartile range) and compared using the Mann–Whitney U test. Non-parametric tests were used if the data were not normally distributed. A value of *p* < 0.05 was accepted as indicative of statistical significance. The relationship between FFR and δdP_n_/dt was quantified with a coefficient of determination (r^2^). The performance of δdP_n_/dt was assessed using sensitivity, specificity, positive predictive value (PPV), negative predictive value (NPV) and diagnostic accuracy (the percentage of patients correctly diagnosed by δdP_n_/dt), together with their 95% confidence intervals (CIs). δdP_n_/dt was compared with the FFR using the receiver operating characteristics (ROC) area under curve (AUC) analyses. The values were defined as significant when the *p* value was <0.05.

## 3. Results

Altogether 232 measurements were performed, of which 116 were carried out under hyperaemic conditions and 116 were carried out under resting conditions. The average FFR value was 0.80 [0.73–0.84]. Here, 88% of the stenoses lay in the 0.60–0.80 FFR range. δdP_n_/dt of the intracoronary pressure trace decreased significantly from rest to the hyperaemic phase (6.31 ± 2.29; 3.9 ± 1.78, *p* < 0.0001).

Our analysis revealed an excellent correlation between the intracoronary δdP_n_/dt measured during hyperaemia and FFR (r = 0.88, *p* < 0.0001), while the correlation was rather weak when the intracoronary δdP_n_/dt was measured at rest (r = 0.4, *p* < 0.0001) (Figure 3A,B).

However, when examining the relationship between the distal δdP_n_/dt and the resting pressure ratio (RPR), the correlation was better (r = 0.6, *p* < 0.0001; Figure 3C). The diagnostic performance of the distal δdP_n_/dt cut-off value ≤3.52 used to predict a pathological FFR (<0.80) is listed in Table 1. The diagnostic accuracy was 91% vs. 58% for the hyperaemic measurement vs. resting measurement, respectively.

Furthermore, the ROC analysis confirmed that the diagnostic performance of the hyperaemic δdP_n_/dt was better than the resting δdP_n_/dt (AUC = 0.98 vs. 0.74, with an optimal cut-off at 3.52) for distinguishing FFR positive stenoses from FFR negative lesions (see Figure 3D,E).

After demonstrating the strong relationship between FFR and the hyperaemic δdP_n_/dt, the correlation between all the measured δdP_n_/dt values (independent of the hyperaemic or resting condition) was also tested, and the corresponding actual pressure gradient suggested a good relationship (r = 0.85; Figure 3F).

When simulating the effect of flow rate on δdp_n_/dt using a mathematical model of unidimensional blood flow, a good relationship between the pressure ratio (or FFR) and δdp_n_/dt was found (r = 0.98, *p* < 0.0001; Figure 4A). Moreover, in the scatter plot there was good agreement between data values of the theoretical model and the corresponding, experimentally measured δdp_n_/dt values and pressure ratio values (Pd/Pa), respectively (Figure 4B).

## 4. Discussion

In the present study, we demonstrated that the morphological characteristics of the intracoronary recorded dicrotic notch at hyperaemia may provide additional information on the functional severity of coronary stenoses. To the best of our knowledge, this is the first study that demonstrates the close relationship between the quantitative waveform analysis of the dicrotic notch and a pressure gradient not limited to the hyperaemic state.

There are limited data in the literature concerning the behavior of the intracoronary measured dicrotic notch under various physiological circumstances. An early study [14] revealed the benefits of performing a spectral analysis of the dicrotic notch, and the authors assumed that “the lesion may act as a mechanical high-frequency filter”. It should be added that this study examined non-hyperaemic pressure traces, which could explain the observed lower accuracy.

Vavuranakis et al. demonstrated a significant alteration in the pressure waveforms between ostial and distal sites of a mildly diseased coronary artery in patients with normal myocardial perfusion and suggested that these alterations could be influenced by adenosine-induced microcirculatory vasodilatation [15]. In a general agreement with their results [15], we found that hyperaemia reduced the amplitude of the dicrotic notch.

In previous studies, attempts were made to demonstrate that the pressure trace distal to the stenosis carries specific information that indicates the severity of the lesion [16,17]. Some of these observations identified the coronary dicrotic notch as a non-hyperaemic indicator, which may predict the physiologic significance of the coronary artery stenosis [17]. However, these studies did not use any specific algorithm (or dedicated software), they only performed a visual estimation of the pressure trace to analyze the persistence of the dicrotic notch. Previously, our group proposed a quantification method of the pressure waveform by derivation of the pressure traces [18]. In a small patient population, we found a relationship between the characteristics of the non-hyperaemic coronary dicrotic notch and FFR. In the present study we not only confirmed these previous results, but also achieved an even stronger and more accurate correlation while performing the hyperaemic measurements.

There is an ongoing debate on whether a coronary flow parameter measured at rest could be as informative as that assessed during maximum hyperaemia. Our investigation supports the hypothesis that a resting dicrotic notch amplitude is not really suitable for predicting FFR as the δdp_n_/dt measured at rest failed to distinguish FFR ≤ 0.80 to an acceptable diagnostic accuracy. These findings are in accordance with some recently published data [19,20,21,22].

The correlation between the actual pressure ratio and the corresponding time point-related δdp_n_/dt among the measurements both in resting and hyperemic state was almost as high as that between the FFR and the δdp_n_/dt during hyperaemia. This finding could be exploited to improve the quality of the measurement as it offers the possibility of comparing the actual pressure ratio with the actual δdp_n_/dt measured continuously from the beginning of the procedure, especially if the determination of the dicrotic notch amplitude of the derivative curve would be displayed simultaneously with the FFR value.

Noting the close correlation between the actual pressure ratio and the derivative of the dicrotic notch amplitude (i.e., δdp_n_/dt), we suggest that the determination of δdp_n_/dt has the potential to validate the measured pressure ratio, both in the resting state and during hyperaemia. It may be used as a quality control of FFR interpretation while enabling an earlier recognition of possible technical pitfalls of the FFR measurement. Recent data pinpointed the major deceptive effect of signal drift with a possibility of stenosis misclassification [10]. However, no real online solution has been proposed to eliminate the problem, still during the FFR measurement.

In current clinical practice, technical problems of FFR measurement, such as drift of the pressure signal or wedging of the catheter, may remain indeterminate during the FFR assessment. In the case of signal drift, one can only see the problem after the pressure wire has been completely withdrawn into the guiding catheter. In such cases, the whole procedure must be repeated from re-equalizing pressures and re-administering the hyperaemic agent. The wedge position of the fluid-filled catheter at the coronary ostium also gives rise to a signal artefact by damping the aortic pressure signal and disturbing the distal coronary pressure trace.

Both these problems could be detected reliably by analyzing the distal pressure waveform morphology at the dicrotic notch and evaluating the concordance with the δdp_n_/dt and the measured FFR, according to the findings of our study. Signal drift usually produces an apparently low FFR, but this does not lead to a decreased δdp_n_/dt as the waveform is not significantly different from the proximal pressure trace.

However, a slightly wedged catheter position may result in some damping on both the aortic and the distal pressure trace, often without any apparent indication of the problem. In this case, the FFR will be higher compared to that measured with a proper catheter position, but the discordance between the FFR and the calculated δdp_n_/dt may be immediately apparent with a lower-than-expected δdp_n_/dt value.

Overall, the interpretation of the δdp_n_/dt value in parallel with FFR measurement is quite straightforward. In cases of appropriate measurements, the pressure ratio and the δdp_n_/dt value should be in the corresponding range. The discordant values of FFR and δdp_n_/dt represent a technical problem. If FFR displays a significant pressure gradient (<0.8) during maximum hyperaemia and the δdp_n_/dt value is high (>3.52), the distal pressure signal has drifted. However, when a hyperaemic FFR is ≥0.8 but the δdp_n_/dt value is <3.52, one must suspect a dampening of the pressure signal.

## 5. Limitations

One limitation of the study is that the FFR distribution, due to the indication of FFR measurement, was somewhat disproportionate, as most of the patients had an FFR between 0.6 and 0.9, and only a small proportion had severe coronary stenoses with FFR < 0.6 or mild lesions with FFR > 0.9. However, this measured FFR interval is the most common also in real-life clinical practice.

The custom-made software calculated the distal δdp_n_/dt value automatically when the operator positioned the marker to the time-point of FFR measurement; however, the calculation needed manual adjustment in approximately 15% of cases due to a noisy signal. Further investigation is required to prove the advantage of distal pressure analysis with a fully automatic, integrated system, which determines the δdp_n_/dt value together with the corresponding FFR value at the same time in a larger patient population.

## 6. Conclusions

During hyperaemia, the change in the distal pressure waveform corresponds to the actual pressure gradient. Here, δdP_n_/dt is a valuable parameter for the quantification of the waveform. Our measurements confirmed the close correlation between this parameter and the FFR, hence the discordance between them may be a good indicator of possible technical problems of intracoronary pressure measurements, such as signal drift and catheter wedging. Hence, the real-time computation of δdP_n_/dt could be useful in order to avoid such technical artefacts of FFR evaluation.

## Figures and Tables

**Figure 1 jpm-12-02035-f001:**
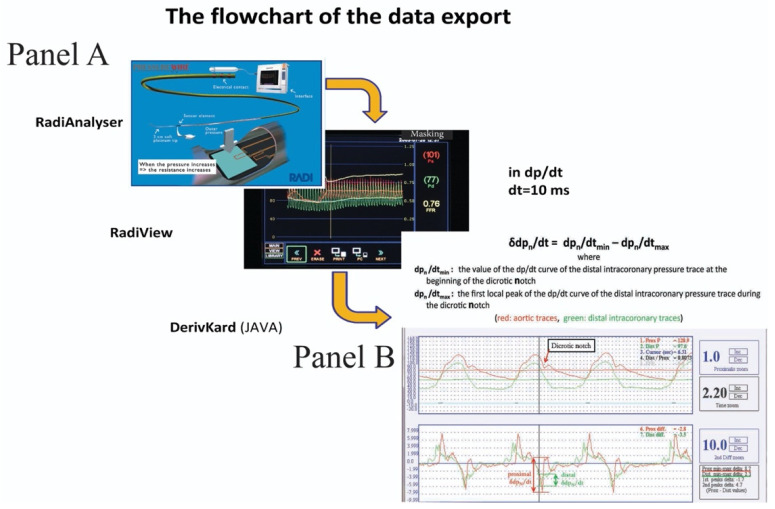
Panel A. Flowchart of data export. Curves were exported from RadiView to our dedicated, custom-made JAVA based software to calculate the derivative curves. Panel B. Calculation of δdP_n_/dt. The upper panel shows the formula for the calculation of δdP_n_/dt. The middle panel shows the original curves recorded without any modification, except for the zoom-in function. This allowed us to identify the exact same time-point with a cursor, where the FFR or RPR calculations were made. The lower panel shows the derivative curves of both the proximal and distal pressure traces with the same time resolution as that of the original curves. Abbreviations: FFR = Fractional flow reserve; δdP_n_/dt = local amplitude around the dicrotic notch on the derivative curve of the distal intracoronary pressure trace; RPR = Resting pressure ratio.

**Figure 2 jpm-12-02035-f002:**
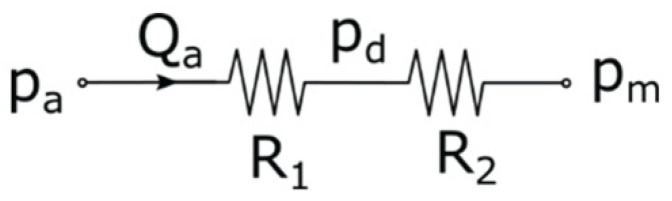
In the unidimensional model, the stenosis and the arterioles are represented by two resistors (R_1_ and R_2_) connected in series.

**Figure 3 jpm-12-02035-f003:**
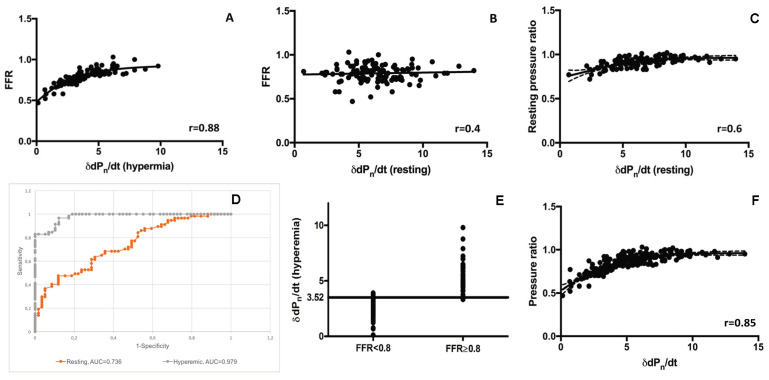
Relationship of FFR and RPR vs. δdP_n_/dt and diagnostic characteristics of δdP_n_/dt using FFR as a reference standard. (**A**,**B**). Scatter plots showing FFR as a function of the hyperaemic δdP_n_/dt (**A**) and that of δdP_n_/dt measured at rest (**B**). Here, r is calculated by applying Spearman’s correlation. (**C**). A scatter plot showing RPR as a function of the distal δdP_n_/dt measured at rest. Here, r is calculated by applying Spearman’s correlation. (**D**). Receiver operating characteristic curve (ROC) for hyperaemic and resting δdP_n_/dt vs. FFR. (**E**). Relationship between individual FFR values grouped into positive (<0.8) and negative (≥0.8) subsets and δdP_n_/dt (hyperaemic). (**F**). Scatter plots showing the phase pressure ratio as a function of δdP_n_/dt. Here, r is calculated by applying Spearman’s correlation. Abbreviations: FFR = Fractional flow reserve; δdP_n_/dt = local amplitude around the dicrotic notch on the derivative curve of the distal intracoronary pressure trace; RPR = Resting pressure ratio; ROC = receiver operating curve.

**Figure 4 jpm-12-02035-f004:**
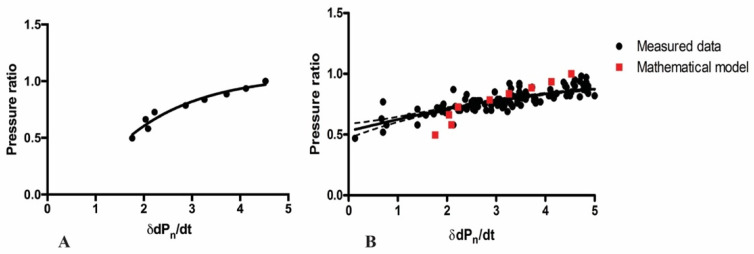
(**A**). Correlation between the δdp_n_/dt and the pressure ratios (Pd/Pa) in the unidimensional simulation at different flow rates. The original (100%) flow rate was 4.66 mL/s. (**B**). shows an identical part of scatter plot data of the correlation between all the measured pressure ratios and δdp_n_/dt (black) and that of data deriving from the mathematical model (red). In this setting, the general scatter of the modeled data and measured data is quite similar. Abbreviations: δdP_n_/dt = local amplitude around the dicrotic notch on the derivative curve of the distal intracoronary pressure trace; Pd/Pa = pressure ratio.

**Table 1 jpm-12-02035-t001:** Diagnostic performance and accuracy of δdP_n_/dt ≤ 3.52.

	% of FFR ≤ 0.80 (*n*)	% of Sensitivity (95% CI)	% of Specificity (95% CI)	% of PPV (95% CI)	% of NPV (95% CI)	% of Accuracy
Hyperaemic (*n* = 116)	50 (58)	90 (79–96)	91 (81–97)	91 (82–96)	90 (80–95)	91
Resting (*n* = 116)	50 (58)	19 (10–31)	98 (91–100)	92 (59–99)	54 (51–57)	58

CI—confidence interval; PPV—positive predictive value; NPV—negative predictive value.

## Data Availability

The data that supports the findings of this study are available with the corresponding author.

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
