# Peer review of "Novel Method to Detect Pitfalls of Intracoronary Pressure Measurements by Pressure Waveform Analysis"

_jpm, 2022, doi:10.3390/jpm12122035_

Round 1
Reviewer 1 Report
Dear Authors,
I congratulate to this meticulous analysis and to the enormous and successful work developing this custom-made software which involves high-quality collaboration between IT and medical experts.
Minor notes: In the title and abstract you shorten the "take-home message" as "quality control". I recommend to rephrase this like: Novel, real-time method to detect inaccuracy of pressure recording during FFR measurements". In the Introduction I would use "decision for revascularization"instead of "PCI outcome". Table 1 is not necessary.
You correctly state in the limitation section that FFR distribution was not proportionate. Please, elaborate whether there is a need for validation of your findings in those "strongly" negative or abnormal situations.
Sincerely yours
Author Response
Thank you very much for your useful suggestions and comments.
The advice for a new title is extremely valuable, thus we changed the title of the manuscript as follows:
Novel method to detect pitfalls of intracoronary pressure measurements by pressure waveform analysis.
The reason we did not include “real time” in the title is the fact that it is rather the aim of this method, however, in the submitted paper it has not been fully implemented yet. Therefore, it is only mentioned as an achievable target.
However, we added the following phrase to the abstract:
Hence, a real time analysis of the ddPn/dt may be helpful in avoiding some pitfalls of FFR evaluation.
We changed the “decision for revascularization” term in the introduction to the following sentence:
Despite the fact that the FFR measurement improves the appropriateness of the decision for revascularization…
According to the Reviewer’s recommendation, we omitted Table 1.
Regarding the disproportionate FFR distribution, we do not feel it would constitute a real drawback of the study, as our patient population is also involving some cases with an FFR between 0.6 and 0.9, in which subgroup the results are in line with the general correlation. On the other hand, the mathematical model has confirmed the relation in a proportionate dataset.

Reviewer 2 Report
Coronary angiography is the standard examination for the evaluation of coronary stenoses. This allows assessment of the severity of the stenosis, but does not provide details about its actual impact on the blood flow. For this reason, the fractional flow reserve (FFR), which represents the average distal-to-proximal pressure ratio in a coronary stenosis under maximal hyperemia, was developed. FFR is an effective index in identifying coronary stenoses that decrease myocardial blood perfusion and produce myocardial ischemia. FFR is measured under maximal hyperemia induced by means of a vasodilatation on the coronary microvasculature. Arterial stenoses acting as filters can attenuate the transmission of pressure waves towards the distal vessel, reducing the components of the high frequency waves with the disappearance of the dicrotic notch.
The presented study demonstrated that the morphology of the intracoronary pressure wave, recorded during hyperemia, with the disappearance of the dicrotic notch, can provide additional information regarding the functional severity of the coronary stenoses. During hyperemia, the change of waveform of the distal pressure corresponds to the real pressure gradient. The distal intracoronary pressure gradient is a valuable parameter for waveform quantification. The results of the study have confirmed the close correlation between this parameter and FFR, specifying also the fact that the discordance between them can be a good indicator of the possible technical problems of intracoronary pressure measurements. Real-time calculation of distal intracoronary pressure gradient could be useful for avoiding technical artifacts of FFR assessment.
The article is well written and documented. I recommend it for publication.
Author Response
Response:
We are grateful for the Reviewer’s thorough evaluation.
Reviewer 3 Report
Well performed study. On my FFR measurement, usually I don't have drift more than 2 mmHg on pull back in the end. However, this is a good method to assess for discrepancy.
Author Response
Response:
We thank the Reviewer for a positive opinion and comment. We completely agree that the contemporary pressure wires have less drift than those of previous generation. However, even a small drift may cause diagnostic uncertainty at the cut-off value of the FFR. On the other hand, a small wedging of the catheter (with minor dumping on the pressure trace) is not always detectable during the measurement already. In our opinion, this method may help to reveal such discrepancies. (We added an illustrative case of wedged pressure traces to the central illustration.)